# Is It Time for a Paradigm Shift in the Treatment of Schizophrenia? The Use of Inflammation-Reducing and Neuroprotective Drugs—A Review

**DOI:** 10.3390/brainsci13060957

**Published:** 2023-06-15

**Authors:** Antonino Messina, Carmen Concerto, Alessandro Rodolico, Antonino Petralia, Filippo Caraci, Maria Salvina Signorelli

**Affiliations:** 1Department of Clinical and Experimental Medicine, Institute of Psychiatry, University of Catania, 95123 Catania, Italy; carmenconcerto@hotmail.it (C.C.); alessandro.rodolico@me.com (A.R.); petralia@unict.it (A.P.); maria.signorelli@unict.it (M.S.S.); 2Department of Drug and Health Sciences, University of Catania, 95123 Catania, Italy; fcaraci@unict.it; 3Unit of Translational Neuropharmacology and Translational Neurosciences, Oasi Research Institute-IRCCS, 94018 Troina, Italy

**Keywords:** schizophrenia, resistant schizophrenia, ultra-high-risk psychosis, neuroinflammation, cytokines, IL-6, TNF-α, IL-1β, anti-inflammatory drugs, neuroprotective drugs

## Abstract

Comprehending the pathogenesis of schizophrenia represents a challenge for global mental health. To date, although it is evident that alterations in dopaminergic, serotonergic, and glutamatergic neurotransmission underlie the clinical expressiveness of the disease, neuronal disconnections represent only an epiphenomenon. In recent years, several clinical studies have converged on the hypothesis of microglia hyperactivation and a consequent neuroinflammatory state as a pathogenic substrate of schizophrenia. Prenatal, perinatal, and postnatal factors can cause microglia to switch from M2 anti-inflammatory to M1 pro-inflammatory states. A continuous mild neuroinflammatory state progressively leads to neuronal loss, a reduction in dendritic spines, and myelin degeneration. The augmentation of drugs that reduce neuroinflammation to antipsychotics could be an effective therapeutic modality in managing schizophrenia. This review will consider studies in which drugs with anti-inflammatory and neuroprotective properties have been used in addition to antipsychotic treatment in patients with schizophrenia.

## 1. Introduction

Within psychotic disorders, schizophrenia represents a chronic illness with a poor outcome. Identifying the causes and treatment of severe psychiatric illnesses, such as schizophrenia, is a challenge for healthcare systems worldwide, as patients with severe mental disorders have a higher mortality rate than the general population [1]. Although atypical antipsychotics are effective in controlling the symptomatology of schizophrenia, there are no drugs to date that can impact the pathogenetic core of schizophrenia, which appears uncertain. Above all, it is a challenge to treat patients with resistant schizophrenia, i.e., the condition in which two antipsychotic drug trials have failed to cause remission. In these cases, the only effective and available drug remains clozapine [2]. Schizophrenia presents positive symptoms such as delusions and hallucinations, and negative symptoms such as poor thoughts, flat affect, apathy, social withdrawal, and cognitive and disorganized symptoms [3]. Although the causes of schizophrenia remain unclear, there is a growing interest in exploring the neuroinflammatory and immune hypothesis as a potential contributor to the disorder’s pathophysiology [4]. Mediators of neuroinflammation are cytokines that are also implicated in neurons’ generation, differentiation, and maturation. Cytokine levels under physiological conditions fluctuate at specific periods when significant changes occur in the prefrontal cortex. Specifically, a peak of Interleukins (IL) occurs in pre-school age and another peak of tumor necrosis factor-α (TNF-α) and Interleukin-6 (IL-6) in adolescence [5].

Several noxious stimuli can trigger cytokines production by microglia (Table 1), which can switch from an anti-inflammatory M2 phenotype to an M1 phenotype that fuels the neuroinflammatory process [6]. Various risk factors are correlated with neuroinflammation (Table 1). Prenatal (i.e., maternal immune activation, MIA, caused by infections during pregnancy), perinatal (i.e., hypoxia at birth) [7], and postnatal stimuli (trauma, stress, infections) can increase the immune system’s reactivity, which, through cytokines production, produces hyperactivation of microglia and, in the long run, neuronal damage, neurotransmission abnormalities, and neurodegeneration [8]. Patients with schizophrenia show higher levels of proinflammatory cytokines such as Interleukin-1β (IL-1β), IL-6 in the blood and cerebrospinal fluid of individuals with the disorder, TNF-α, and as well as increased activation of microglia and astrocytes and an unspecific inflammatory blood marker, the C-reactive protein (CRP) [9,10]. Particularly at the onset of schizophrenia and during the recurrence of psychotic episodes, blood levels of proinflammatory cytokines such as IL-1β, IL-6, and TNF-α tend to increase [11], and levels of IL-6 are associated with poor schizophrenia prognosis [12]. In fact, microglia activation during psychotic relapses results in an increased production of proinflammatory cytokines [11]. Intriguing preclinical research found that animals exposed to MIA showed an increased expression of the nod-like receptor pyrin domain-containing protein 3 (NLRP3) inflammasome. The high expression of NLPR3, which is involved in the inflammatory pathway, is associated with schizophrenia-like behavior [13]. Moreover, the activation of microglia induces increased oxidative stress related to the increase in quinolinic acid produced by the kynurenic acid pathway [14]. Indeed, tryptophan metabolism, in which the kynurenine pathway is involved, is impaired in schizophrenia [15]. Thus, high levels of kynurenic acid can damage dopaminergic and glutamatergic neurotransmission and lead to psychotic symptoms and cognitive impairment [16,17]. Furthermore, the neuroinflammation-related state of microglia activation leads to a reduction in brain-derived neurotrophic factor (BDNF), with neuron loss, reduced synaptic plasticity, and consequent neurodegeneration [18].

The gene expression of DNA sequences coding for proteins involved in TNF-α and IL-17 signaling processes appears more pronounced in schizophrenia patients than in healthy ones [19]. Multiple genes, such as FOS, IL1B, CXCL8, CASP1, CFL1, CAMP, ITPR2, and ACTG1, implicated in immune response and inflammation, are more highly expressed in schizophrenia than in the general population [19].

In light of this, a paradigm shift has been taking place in recent years regarding psychiatric disorders. Emerging evidence brings schizophrenia closer to multiple sclerosis relative to the pathogenetic basis, albeit with different anatomopathological and clinical manifestations [20]. Whether the pathogenetic processes of schizophrenia are similar to the neuroinflammation observed in multiple sclerosis, several clinical trials have investigated the role of anti-inflammatory and immunomodulatory therapies in treating schizophrenia. This finding is consistent with risperidone’s efficacy in reducing the blood concentration of IL-6, which appears higher in patients with schizophrenia than in controls [21]. Recently, lumateperone, an atypical antipsychotic, which modulates dopaminergic, serotoninergic, and glutamatergic neurotransmission, has been shown to have anti-inflammatory activity, reducing the levels of IL-1β, IL-6, and TNF-α and promoting the restoring of blood–brain barrier (BBB) integrity [22]. In a study conducted by Fitton [23], the researchers examined the potential use of anti-inflammatory medication to treat mental disorders. Their review involved analyzing existing literature, specifically emphasizing controlled trials and systematic reviews.

The treatment of schizophrenia is a challenge for the clinician. A meta-analysis of 62 double-blind, randomized studies showed that different molecules with anti-inflammatory action improved both positive and negative symptoms of schizophrenia [24]. Given these findings, researchers have investigated the use of anti-inflammatory and monoclonal antibody drugs as promising add-on treatments for schizophrenia. Due to their anti-inflammatory properties and neuroprotective action [25], these drugs belong to different pharmacological categories and can be defined as neuroinflammatory-reducing and neuroprotective drugs (NRNDs). These drugs alleviate neuroinflammation, show a demonstrated neuroprotective effect, and improve symptoms in patients with schizophrenia. As much as treatments with second- and third-generation atypical antipsychotics are valuable tools in terms of efficacy and tolerability, they do not affect the pathogenetic mechanisms of schizophrenia, but rather the epiphenomena represented by neurotransmission abnormalities.

This systematic search followed the PRISMA guidelines. Two authors independently searched the MEDLINE, Cochrane Central Register, EMBASE, and Mendeley databases for the following entries: schizophrenia or patients with schizophrenia and celecoxib and PUFA and omega-3-fatty acids and acetylsalicylic acid and minocycline and statins and PPAR agonist and pioglitazone and rosiglitazone and ace-inhibitors and prednisolone and immunomodulators and fingolimod and monoclonal antibody and rituximab. Only English-written papers were considered (Figure 1).

## 2. Polyunsaturated Fatty Acids

Omega-3 and omega-6 fatty acids belong to polyunsaturated fatty acids (PUFAs). PUFAs are essential constituents of neuronal membranes and provide proper membrane function. A large cross-national study showed a correlation between low levels of PUFAs and an increased risk of schizophrenia [26]. PUFAs reduce neuroinflammation and, in patients with schizophrenia, result in a reduction in proinflammatory cytokines, such as IL-6 and TNF-α; a reduction in CRP; and an increase in BDNF, which, due to its neurotrophic action, has positive effects on cognition function [27]. Given omega-3 fatty acids’ neuroprotective and antioxidant effects, they have been proposed as a treatment in first-episode psychotic patients or in ultra-high-risk state subjects (UHR), i.e., those with subthreshold psychotic symptoms at risk of developing full-blown psychosis [28,29,30,31]. In addition, some authors found that the reduction in omega-3 fatty acid in the erythrocyte membrane (omega-3 index) could be a biomarker of risk in UHR individuals [32] and a risk factor for drug treatment resistance [33]. The efficacy and safety of PUFAs augmentation to antipsychotic therapy have been demonstrated in a meta-analysis of randomized controlled trials [34]. Another meta-analysis of RCTs reported that the assumption of 1 g/day of omega-3 fatty acid improved positive symptomatology [35]. Some authors in a randomized clinical trial (RCT) reported a reduction in violent behavior in patients with schizophrenia treated with PUFAs at twelve weeks [36]. However, another meta-analysis concluded that although there are some efficacy data, these are of poor quality, and further studies would be needed [37].

## 3. Statins

Like other molecules, statins, drugs that inhibit 3-hydroxy-3-methyl-glutaryl-coenzyme A (HMG-CoA) reductase by inducing a lowering of cholesterol levels, used in hypercholesterolemia, also possess anti-inflammatory activity [38]. Statins can be distinguished into hydrophilic: pravastatin and rosuvastatin; and lipophilic: atorvastatin, fluvastatin, lovastatin, pitavastatin, and simvastatin [39]. Two meta-analyses, which included six randomized clinical trials (RCTs), observed in patients taking statins in addition to antipsychotics, showed a reduction in positive and negative symptoms, compared with the control group not taking them [40,41]. At a daily dose of 40 mg, simvastatin added to risperidone proved effective in reducing negative symptoms of schizophrenia at eight weeks but did not show the same effectiveness in controlling positive symptoms [42]. Nevertheless, not all studies agree: some authors came to opposite conclusions of no efficacy [43,44], and in a meta-analysis, statins were not reported to be effective in controlling the severity of schizophrenia symptoms, regardless of the molecule’s tendency to pass the BBB [18]. In a large retrospective study performed on veterans with schizophrenia, the authors observed that the risk of incurring hospitalization was lower in patients taking statins [45]. The effect of statins may be due to their effect in reducing neuroinflammation [46], and decreasing blood values of IL-1β, IL-6, TNF-α, and C-reactive protein (CRP) [46,47]. Within the prefrontal cortex of patients with schizophrenia, the gene expression of the Toll-like receptors 4 (TLR4), pivotal in the proinflammatory pathway, is altered [48]. In schizophrenia, statins have been shown to effectively modulate both NLRP3 inflammasome and TLR pathways involved in neuroinflammation [47]. However, it is necessary to remember that not all statins are the same: some are lipophilic, while others are hydrophilic. Lipophilicity ensures their passage through the blood–brain barrier (BBB); thus, in studies involving these drugs, one must keep this in mind, as some may be biased by the ineffectiveness of statins that do not pass the BBB.

## 4. Peroxisome Proliferator-Activated Receptors’ Agonists

The peroxisome proliferator-activated receptors (PPARs) are intranuclear receptors, which act as transcription factors, binding to DNA and thus regulating gene expression [49]. In inflammatory processes, a key role is played by the nuclear factor kappa-light-chain-enhancer of activated B cells (NFkB), a transcription factor that stimulates the expression of enzymes involved in the prostaglandin pathway by inducing COX-2 gene expression [50]. The pro-inflammatory action of NFkB is inhibited by the PPARs, which comprise three isoforms, PPAR-α, PPAR-β/δ, and PPAR-γ [50]. PPAR-γ is widely expressed in microglia and exhibits a potent anti-inflammatory activity, influencing multiple pathways through inhibiting cytokine gene expression and prostaglandins and inducing apoptosis in activated microglia cells [51]. On the other hand, the main effect of PPAR-α is to facilitate neurotransmission processes and have a neuroprotective effect, while the action of PPAR-β/δ is unknown [52].

Because NFkB and PPARs are dysregulated in schizophrenia and are associated with higher levels of neuroinflammation [50], the agonist of PPARs can reduce inflammatory processes, reducing TNF-α and IL-6 levels [50,53]. PPARs not only inhibit NFkB gene expression, but also modulate the action of TLRs, which, as already mentioned, play a key role in the production of proinflammatory cytokines and the triggering of the neuroinflammatory process [54]. The PPARs agonists approved to date for the treatment of diabetes are rosiglitazone and pioglitazone. In preclinical studies, rosiglitazone improved memory because of its positive effect on BDNF gene expression [55].

The use of pioglitazone has been studied in patients with schizophrenia, and at a dosage of 30 mg per day for eight weeks resulted in a reduction in the severity of symptoms of the disorder [18]. It would also appear that pioglitazone, in addition to antipsychotics, improves negative symptomatology [56]. In view of the broad action of PPARs in neurons, it would be opportune to investigate PPARs agonists extensively, as also suggested in a recent review on the potential use of PPARs agonists in psychopharmacology [57].

## 5. AT1 Antagonists and ACE Inhibitors

Interestingly, the renin–angiotensin system (RAS) and angiotensin-converting enzyme (ACE), primarily involved in blood pressure regulation, appear to modulate PPARs and neuroinflammation and regulate GABAergic and dopaminergic neurotransmission, which are involved in schizophrenia [58,59]. According to recent evidence, RAS and ACE appear to be linked to neurodegenerative diseases and schizophrenia [60], and reduced ACE levels have been found in patients with schizophrenia [61]. Thus, using drugs that modulate RAS, such as angiotensin 1 receptor (AT1) antagonists and angiotensin-converting enzyme (ACE) inhibitors, could help treat the neuroinflammatory processes underlying the pathogenesis of schizophrenia. The pleiotropic activity of AT1 antagonists, which contributes to reducing neuroinflammation, modulating the immune response and the coagulation cascade, and protecting endothelial cells and mitochondria, can explain the role of AT1 antagonists in preventing neurodegeneration observed in schizophrenia [62]. The anti-inflammatory properties of AT1 antagonists are likely to be related to the decrease in pro-inflammatory cytokines, mediated by the reduction in gene expression of NLPR3 and NF-κB [62]. The disruption of the BBB, which is related to neuroinflammation, is increased by AT1 receptors, so the use of AT1 antagonists lowers the permeability of the BBB, thereby preventing harmful agents from penetrating the brain [62,63].

Telmisartan, an AT1 antagonist, has been shown to effectively reduce the neurotoxic effect of IL-1β that can result in neurodegeneration [64]. Moreover, the use of telmisartan appears to be efficacious, in addition to clozapine or olanzapine, in improving the symptomatology of schizophrenia [65].

Preclinical studies observed that AT1 antagonists, particularly irbesartan, losartan, and telmisartan, reduce levels of kynurenic acid, which at high levels results in the blockade of NMDA glutamate receptors, associated to the onset of psychotic symptoms [66]. In mice models, candesartan reduces hippocampal microglia activation [67].

Moreover, ACE inhibitors alter the metabolism of kynurenic acid. In vitro studies on rat cortex showed that among the various ace inhibitors, while lisinopril tends to increase kynurenic acid levels, ramipril conversely reduces them. In contrast, perindopril appears to have a neutral action on kynurenic acid levels [68].

## 6. Acetylsalicylic Acid and Other Nonsteroidal Anti-Inflammatory Drugs

Acetylsalicylic acid, a non-selective COX inhibitor, modulates cy-cyclooxygenase-2 (COX-2) and inhibits cyclooxygenase-1 (COX-1) irreversibly. The anti-inflammatory action of acetylsalicylic acid is achieved by inhibiting the production of thromboxanes and prostaglandins [69] and has proven effective in addition to antipsychotic therapy in reducing both positive and negative symptoms of schizophrenia [18,70,71]. The dosage of acetylsalicylic acid used in patients with schizophrenia ranged from 325 mg up to 1000 mg daily. Acetylsalicylic acid is effective in reducing the production of IL-6 and TNF-α and protecting against oxidative stress damage [72]. In a meta-analysis that considered different nonsteroidal anti-inflammatory drugs (NSAIDs), such as ibuprofen, diclofenac, naproxen sodium, and acetylsalicylic acid, it was observed that the augmentation of NSAIDs to antipsychotics was effective in reducing the severity of symptoms of schizophrenia [73].

## 7. Celecoxib

Celecoxib, a drug that inhibits the enzyme cyclooxygenase-2 (COX-2), has been investigated as an additional treatment option for schizophrenia. COX-2, unlike the other isoform of the enzyme COX-1, plays a specific role in the pathogenesis of inflammation [74]. COX-2 is also expressed in nervous tissue, and through the production of prostaglandin E2 modulates immune action in the central nervous system (CNS) and plays a crucial role in neuroinflammatory processes [75], with specific involvement of the hippocampus as well [76]. Various researchers have reviewed randomized clinical trials that assessed using celecoxib as an add-on treatment for schizophrenia [77]. The action of celecoxib manifests through its neuroprotective and immunomodulatory effects [78]. In a double-blind study, the combination of 400 mg/day of celecoxib with risperidone at standard dosages (2–6 mg/day), regardless of sex, age, and duration of illness, was more effective in improving positive and negative symptomatology in schizophrenia [79,80]; the same effect was observed with the association of celecoxib and amisulpride [81]. The cognitive function of patients with schizophrenia also improved following the addition of celecoxib [82]. However, other authors using 400 mg/day of celecoxib combined with an antipsychotic found no difference from using an antipsychotic [83]. It is likely that the effectiveness of celecoxib would depend on the stage of schizophrenia, being more useful in the early rather than later stages [77,84]. This datum is confirmed by a meta-analysis that concluded that further use of celecoxib is more effective in the first episode of schizophrenia [85].

## 8. Minocycline

Minocycline, a tetracycline antibiotic, has been investigated for its potential anti-inflammatory effects in treating schizophrenia. Preclinical studies in mice have shown that minocycline can reduce microglia activation at the hippocampal and prefrontal levels [18,86]. Many authors have conducted meta-analyses of randomized controlled trials [87,88,89,90]. They found that minocycline significantly improved negative symptoms of schizophrenia and general psychopathology and reduced inflammation markers, especially in studies where the treatment lasted longer. However, the authors did not report differences regarding positive symptoms. Minocycline combined with clozapine was an optimal treatment strategy in resistant schizophrenia. The treatment’s efficacy in the add-on can also be an effect of increased clozapine plasma levels caused by minocycline [91]. Specifically, in resistant patients with schizophrenia, improvement was mostly observed in cognitive function and in reducing avolition [92]. These findings suggest that minocycline may be a promising additional treatment for schizophrenia, particularly for patients experiencing cognitive impairment and negative symptoms [93,94,95,96]. The improvement of cognitive function was associated with a reduction in a marker of neuroinflammation interleukin-6 [97], and greater efficacy of minocycline appears to be related to higher neuroinflammation [98]. Due to its neuroprotective and anti-inflammatory properties, minocycline reduces microglia activation observed in patients with schizophrenia [99]. Minocycline may facilitate the transition from M1 to M2 by inhibiting microglia hyperactivation and related neuroinflammation [100]. The inactivation of microglia, and consequently the reduction in IL-1β, IL-6, and TNF-α levels, is mediated by the suppression of TLR4 signaling [101].

In addition to its effects on microglia, minocycline exerts neuroprotective and antiapoptotic actions [102]. The hippocampus, a formation involved in schizophrenia, is one of the targets of minocycline, which stimulates neurogenesis and reduces microglia activation [103]. Exciting speculation hypothesizes that minocycline acts on microglia and regulates the remodeling synapses and circuits involved in the “social brain” [104]. Several pieces of evidence have shown that the NMDA glutamate receptor plays a key role in the pathogenesis of schizophrenia [105]; in fact, molecules that antagonize the NMDA receptor cause the onset of psychotic symptoms. Minocycline inhibits the neurotoxicity of NMDA receptor antagonists [106].

As with other molecules considered in add on, there are conflicting studies for minocycline. In RCTs, the authors found no difference between patients taking an antipsychotic and minocycline at 200 mg/day and the group taking a placebo [107,108,109]. Nevertheless, given the amount of positive data on the use of minocycline and considering some conflicting data to date, it cannot be completely ruled out that minocycline may find efficacy in patients with the positivity of inflammatory biomarkers. Some studies do not consider patients with more severe symptoms, relapses, and the presence of negative symptoms before the start of minocycline treatment [110]. More homogeneous studies by disease duration and severity, differentiated by symptom cluster, and considering neuroinflammatory profile would be needed to clarify the usefulness of minocycline in the treatment of schizophrenia in combination with antipsychotics.

## 9. Prednisolone

Prednisolone, a corticosteroid, has also been studied as an adjunctive treatment for schizophrenia. Nitta [111] conducted a meta-analysis of randomized controlled trials investigating prednisolone use in schizophrenia and found some evidence for its effectiveness. However, the study had methodological limitations, and the overall effect size was small. Nevertheless, it must be considered that high cortisol levels are associated with psychotic symptoms, so the use of prednisolone may be risky [112].

## 10. Immunomodulator Drugs

The new frontier in treatment studies of schizophrenia is the use of immunomodulators. Among them, fingolimod, used in treating multiple sclerosis, which possesses marked anti-inflammatory and neuroprotective activity, appears effective in improving the cognitive symptoms of schizophrenia. Preclinical studies showed that fingolimod reduces microglial activation and levels of proinflammatory cytokines such as IL-6, while increasing BDNF [113]. The effect of fingolimod is expressed in the increase in white matter at the level of the corpus callosum and superior longitudinal fasciculus, and the reduction in lymphocyte counts [114]. The protective action of fingolimod would be related to the direct effect of the molecule on oligodendrocytes [115]. In an RCT, some authors found a significative improvement in negative symptomatology and global functioning in 80 patients with schizophrenia taking fingolimod, compared with as many patients taking a placebo [116].

Another drug used to treat rheumatoid arthritis is methotrexate. This drug, used once weekly at 10 mg, effectively reduces positive symptoms while remaining ineffective on negative ones [117]. However, methotrexate, which has antagonistic effects on folic acid synthesis, is burdened by severe side effects on the immune system that make it hardly usable.

## 11. Monoclonal Antibodies

According to emerging studies, monoclonal antibodies may also play a role in treating some psychopathological domains of schizophrenia. Monoclonal antibodies are a class of molecules that act by antagonizing the cytokines. The main field of use of this category of drugs is oncological disease. Several monoclonal antibodies exist, among which adalimumab has proven to be significantly superior to placebo in combination with risperidone in treating the negative symptoms of schizophrenia [118]. Among the many molecules, adalimumab selectively binds to TNF-α by preventing its action on the receptor [119]. Another piece of research showed an improvement in the general symptomatology of schizophrenia with efficacy in improving global functioning in a small group of resistant patients treated with rituximab [120], which targets CD20, a transmembrane protein present on B lymphocytes whose proliferation it inhibits. Cognitive improvement was observed with the administration of tocilizumab, an IL-6 antagonist [121]; this datum was not confirmed in trial, however, which did not attribute the ineffectiveness to the molecule itself, but to the fact that tocilizumab passes BBB with difficulty [122]. In a recent review, the investigation of the efficacy of rituximab and ocrelizumab on the cognitive function of patients with schizophrenia yielded controversial results. However, the use of adalimumab has been shown to be effective in controlling negative and positive symptoms of schizophrenia [123].

## 12. Conclusions

Although the use of anti-inflammatory drugs as supplementary treatments for schizophrenia shows potential, more research is necessary to determine their ideal usage and safety. According to the neuroinflammatory hypothesis of schizophrenia, inflammation plays a critical role in the etiology and neuro-progression of the disorder. Thus, neuroinflammation-reducing and neuroprotective drugs (NRNDs) hold promise as a potential treatment option. However, the complexity of schizophrenia and the interaction between inflammation and other biological and psychosocial factors make it challenging to identify patient groups that could benefit from NRNDs (Table 2). Therefore, future research should strive to identify biomarkers that could aid in predicting treatment response and explore the optimal dosing and duration. NRNDs could be used as an add-on to antipsychotics in some forms of schizophrenia in which the neuroinflammatory component is more significant, or in predominantly negative or cognitively impaired schizophrenia, in resistant form, and specific internist comorbidity (Table 2 and Table 3). In this regard, in patients with schizophrenia, it would be desirable for neuroinflammatory screening to be carried out, allowing patients with neuroinflammatory schizophrenia to be identified and treated appropriately.

In this regard, to improve and individualize the pharmacological treatment of schizophrenia, some authors have proposed using pro-inflammatory cytokines as a biomarker to stage schizophrenia from the prodromal stages, the first episode, to chronic forms in relation also to the predominance of negative or positive symptoms [124]. Patients with treatment-resistant forms of schizophrenia, who account for 30% of patients with schizophrenia, could benefit from a staging involving pro-inflammatory cytokines dosage to tailor therapy, using drugs that act on neuroinflammatory mechanisms [125].

NRNDs represent a new therapeutic option for patients with schizophrenia. Future research should involve case-control studies differentiated by the subtype of schizophrenia, evaluating the presence of forms with a high neuroinflammatory component versus forms of schizophrenia with a low neuroinflammatory component, as inferred from serological biomarkers. In addition, the use of NRNDs should be investigated in schizophrenia variants with the prevalence of negative or positive symptomatology and a possible impact on cognitive function.

## Figures and Tables

**Figure 1 brainsci-13-00957-f001:**
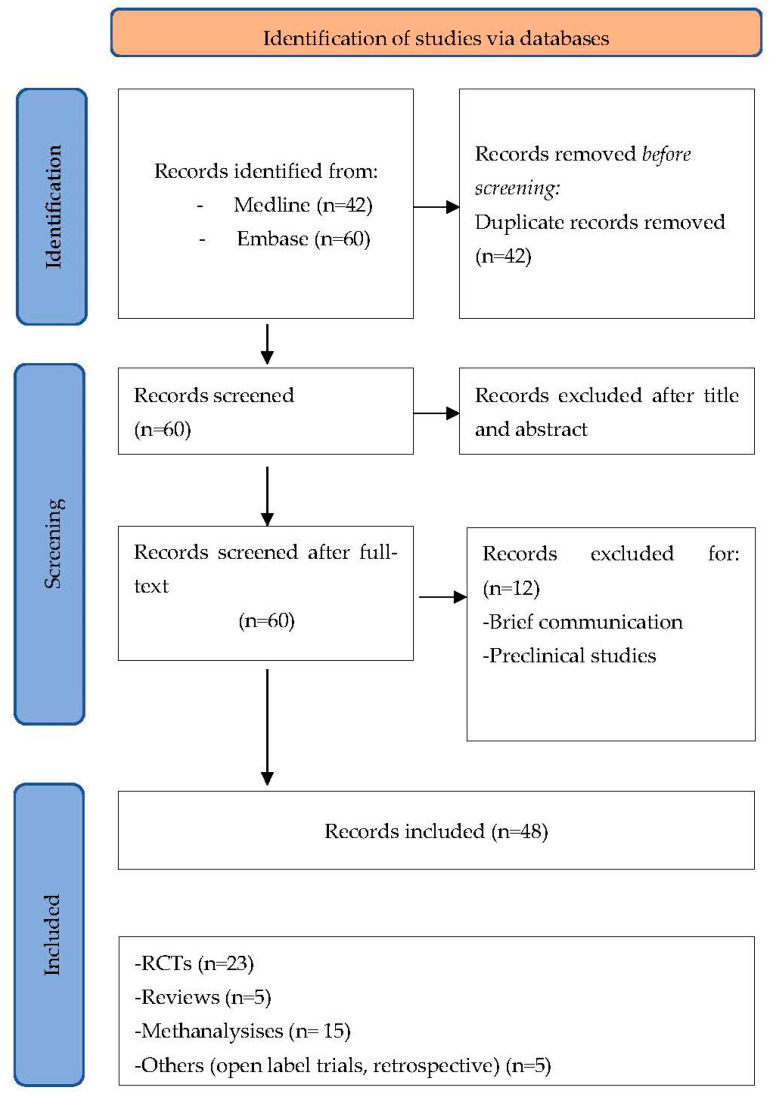
Prisma Flow Diagram.

**Table 1 brainsci-13-00957-t001:** Risk factor of neuroinflammation in schizophrenia.

Risk Factor of Neuroinflammation in Schizophrenia (Hong, 2020 [11])
Infectious agents (i.e., toxoplasma gondii)
Maternal immune activation caused by prenatal infections
Pro-inflammatory genes (FOS, IL1B, CXCL8)
Stress inflammation correlated (psychosocial stress)

**Table 2 brainsci-13-00957-t002:** Possible indications to use NRNDs in schizophrenia.

Early state of psychosis Resistant schizophrenia Schizophrenia with neuroinflammation (CRP, cytokines, L/N ratio) Schizophrenia with prevalent negative and cognitive symptoms PET neuroimaging signs of microgliosis

**Table 3 brainsci-13-00957-t003:** Utilization of NRNDs in different cluster symptoms of the schizophrenia and in relation to the comorbidity.

	Positive Symptoms	Negative Symptoms	Cognitive Symptoms	FEP/UHR	GF	RS	Diabetes	Ch/Tri.	AH	RD
**PUFAs**	√			√		√		√		
**Statins**	√	√						√		
**PPARs agonists**		√					√			
**Minocycline**		√	√			√				
**Celecoxib**	√	√		√						√
**MAb**		√			√	√				
**Fingolimod**		√			√					
**Prednisolone**					√					√
**AT1 antagonists/** **ACE inhibitors**					√	√			√	

MAb: monoclonal antibodies; FEP/UHR: first episode of psychosis/ultra-high risk; GF: global functioning; RS: resistant schizophrenia; Ch/Tri: cholesterol and triglycerides levels; AH: arterial hypertension; RD: rheumatologic diseases. √: drug action on specific symptoms of diseases.

## Data Availability

Suggested Data Availability Statements are available in section “MDPI Research Data Policies” at https://www.mdpi.com/ethics, accessed on 1 July 2023.

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
