# Peer review of "Is It Time for a Paradigm Shift in the Treatment of Schizophrenia? The Use of Inflammation-Reducing and Neuroprotective Drugs—A Review"

_brainsci, 2023, doi:10.3390/brainsci13060957_

Round 1

Reviewer 1 Report

This review consider studies in which drugs with anti-inflammatory and neuroprotective properties have been used in add on to antipsychotic treatment in patients with schizophrenia.the work is a significant contribution to the field of schzophrenia,but in my opinion,The introduction should provide a more detailed explanation of the relationship between neuroinflammation and schizophrenia, such as Zhang et al study found that FOS, IL1B, CXCL8, CASP1, CFL1, CAMP, ITPR2, ACTG1 and other inflammation and immunity related genes were differently expressed between the schizophrenia patient group and the healthy control group, Their findings were consistent withthe neuroinflammatory hypothesis of schizophrenia.(doi:10.12659/MSM.922426).Feng,Z et al found that risperidone can significantly reduce IL-6 levels in schizophrenia. IL-6 is a potential biomarker of the pathophysiology and clinical processes of schizophrenia.(doi: 10.1097/MD.0000000000019694).

 Minor editing of English language required

Author Response

Dear reviewer, thanks for your valuable suggestions. The introduction has been implemented according to your notes. Some insights from Zhang and Feng's papers have been added. A section on genes expressed in schizophrenia has been added.

Reviewer 2 Report

The article titled "Is it time for a paradigm shift in the treatment of schizophrenia? The use of inflammation-reducing and neuroprotective drugs, a review" offers a comprehensive exploration of a potentially groundbreaking approach to treating schizophrenia. Focusing on the potential benefits of inflammation-reducing and neuroprotective drugs, this review article presents a compelling case for a paradigm shift in the field of schizophrenia treatment.

The authors begin by highlighting the current limitations of antipsychotic medications, which have long been the mainstay of treatment for schizophrenia. While these drugs effectively manage some symptoms, they often fall short in addressing the cognitive impairments and negative symptoms associated with the disorder. This leaves a significant gap in the therapeutic options available to patients, highlighting the urgent need for innovative approaches.

The review then delves into the emerging understanding of the role of inflammation in schizophrenia. The authors present a wealth of evidence supporting the hypothesis that neuroinflammation plays a crucial role in the development and progression of the disorder. They explore various inflammatory markers and provide insights into how targeting these inflammatory pathways could potentially alleviate symptoms and halt disease progression.

Furthermore, the article examines the potential of neuroprotective drugs in the treatment of schizophrenia. By focusing on agents that aim to preserve and restore neuronal health, researchers are exploring novel therapeutic avenues. The authors discuss the neuroprotective properties of several drugs and highlight promising preclinical and clinical findings that support their potential efficacy.

What sets this review article apart is its balanced and evidence-based approach. The authors thoroughly analyze a wide range of studies, including preclinical research, clinical trials, and meta-analyses. They present both the successes and limitations of various inflammation-reducing and neuroprotective drugs, providing a comprehensive overview of the current state of research in the field.

The article concludes by emphasizing the need for further investigation and clinical trials to validate the potential of inflammation-reducing and neuroprotective drugs in schizophrenia treatment. It also highlights the importance of personalized medicine and the potential for combining these novel approaches with existing antipsychotic medications for enhanced therapeutic outcomes.

Overall, "Is it time for a paradigm shift in the treatment of schizophrenia? The use of inflammation-reducing and neuroprotective drugs, a review" presents a compelling argument for a new direction in the management of schizophrenia. By focusing on inflammation and neuroprotection, this article offers hope for improved treatment outcomes and represents a significant contribution to the ongoing efforts to enhance the lives of individuals living with schizophrenia.

I suggest that the authors expand the methods section to include a diagram illustrating the PRISMA-Scope-compliant source selection procedure.

Author Response

Dear Reviewer,  We are grateful to you for appreciating our paper. We have gratefully accepted your suggestion and included the PRISMA diagram in the review.

Reviewer 3 Report

This is an interesting topic for review. There are some points for consideration:

This article should include a meta-analysis of the included in the main body of the article studies, otherwise it seems that it is more of a personal opinion. Therefore, it would be necessary to include in the title as well as in the paper that it will be also a meta-analysis.

At the beginning of the article authors should take into consideration relevant already published literature, so they can present a general brief discussion of the increased importance of schizophrenia in psychiatry, e.g. due to the increased mortality cross-culturally for disorders like schizophrenia, which makes the reader see the importance of the the topic under discussion: Giannouli, V. (2017). Ethnicity, mortality, and severe mental illness. The Lancet Psychiatry4(7), 517.

Some more focused articles that authors should read and incorporate are the following:

Hong, J., & Bang, M. (2020). Anti-inflammatory strategies for schizophrenia: a review of evidence for therapeutic applications and drug repurposing. Clinical Psychopharmacology and Neuroscience18(1), 10.

Çakici, N., Van Beveren, N. J. M., Judge-Hundal, G., Koola, M. M., & Sommer, I. E. C. (2019). An update on the efficacy of anti-inflammatory agents for patients with schizophrenia: a meta-analysis. Psychological medicine49(14), 2307-2319.

Pandurangi, A. K., & Buckley, P. F. (2020). Inflammation, antipsychotic drugs, and evidence for effectiveness of anti-inflammatory agents in schizophrenia. Neuroinflammation and Schizophrenia, 227-244.

The tables are easy to read, but there are no conclusions for the readers.

Author contributions at the end are not presented in the relevant section.

Minor language editing is needed throughout the text.

Author Response

Dear reviewer, we are grateful for your suggestions, which we appreciate and consider. The importance of schizophrenia in global health has been discussed more extensively, also mentioning Giannouli. Thank you for also sending us bibliographic suggestions, which we have punctually included in the body of the paper (Hong; Caciki; Pandurangi).
We have revised the tables and enriched them to make them more comprehensive. Finally, the section on authors' contributions has been included.